# Extrusion Foaming of Lightweight Polystyrene Composite Foams with Controllable Cellular Structure for Sound Absorption Application

**DOI:** 10.3390/polym11010106

**Published:** 2019-01-09

**Authors:** Yanpei Fei, Wei Fang, Mingqiang Zhong, Jiangming Jin, Ping Fan, Jintao Yang, Zhengdong Fei, Lixin Xu, Feng Chen

**Affiliations:** 1College of Materials Science and Engineering, Zhejiang University of Technology, Hangzhou 310014, China; 201101391305@zjut.edu.cn (Y.F.); 2111625018@zjut.edu.cn (W.F.); zhongmq@zjut.edu.cn (M.Z.); fanping@zjut.edu.cn (P.F.); yangjt@zjut.edu.cn (J.Y.); feizd@zjut.edu.cn (Z.F.); 2College of Mechanical Engineering, Zhejiang University of Technology, Hangzhou 310014, China

**Keywords:** extrusion foaming, super critical CO_2_, lignin, sound absorption coefficient, mechanical property

## Abstract

Polymer foams are promising for sound absorption applications. In order to process an industrial product, a series of polystyrene (PS) composite foams were prepared by continuous extrusion foaming assisted by supercritical CO_2_. Because the cell size and cell density were the key to determine the sound absorption coefficient at normal incidence, the bio-resource lignin was employed for the first time to control the cellular structure on basis of hetero-nucleation effect. The sound absorption range of the PS/lignin composite foams was corresponding to the cellular structure and lignin content. As a result, the maximum sound absorption coefficient at normal incidence was higher than 0.90. For a comparison, multiwall carbon nanotube (MWCNT) and micro graphite (mGr) particles were also used as the nucleation agent during the foaming process, respectively, which were more effective on the hetero-nucleation effect. The mechanical property and thermal stability of various foams were measured as well. Lignin showed a fire retardant effect in PS composite foam.

## 1. Introduction

With the advancement of industrial modernization, noise pollution has become a worldwide problem affecting the quality of human life. Sound absorption and noise reduction have gradually evolved into a comprehensive subject related to high-tech applications, environmental protection fields, and human coordinated development. The performance and application of novel sound absorption and noise reduction materials have become the development goals of all countries in the world. Compared with traditional metal materials [1], polymer materials [2,3,4,5,6] especially polymer foams [7,8,9,10,11,12], have great advantages in the field of sound absorption due to their light weight and good processability. Sound waves can be greatly absorbed during propagation in foam structures through the internal reflection, refraction and dissipation of the sound waves in a cellular structure. It has been proved that foam structure is the determinant factor affecting the sound absorption efficiency [13]. However, it is difficult in most of the foam preparation methods to control the foam structure and foam density precisely of general polymers (PP, PE, polyolefin) except polystyrene (PS) [14].

PS foam is broadly used in building, transportation, refrigeration, insulation and shock absorbing materials, and occupied the second largest usage of foaming plastics in the market, because of its light weight, low thermal conductivity and good impact resistance [15]. In order to achieve the industrial production and commercial worth of PS foams with low density, low cost and environmental benign process, extrusion foaming by using supercritical CO_2_ (ScCO_2_) as a physical blowing agent has been identified as the most promising technology [16]. However, so far, there have been few reports about controllable cell structure of PS foam for its acoustic property.

In this paper, we demonstrate excellent sound absorption PS composite foams using ScCO_2_ assisted extrusion foaming process. Lignin, a bio-resource material [17,18,19], was employed to control the cell structure and morphology for the first time. The variable foam structure resulted in controllable reflection, dissipation and absorption of sound wave. For comparison, carbonaceous fillers (multiwall carbon nanotube and micro graphite) were invested in for the hetero-nucleation effect on PS foam and its sound absorption property.

## 2. Experimental

### 2.1. Materials and Samples Preparation

Polystyrene (PS) was grade Total 5197 and obtained from the Total Petrochemicals (Houston, Texas, USA). Lignin, micro-graphite (mGr) and multiwall carbon nanotube (MWCNT) were used as the nucleation agent and sound absorption fillers: lignin (kraft lignin, Mw > 5000) was purchased from Aladdin Company (Shanghai, China). And mGr with an average plate diameter of 5 μm and thickness of 0.5 μm were purchased from Qingdao Yanhai Carbon Materials Inc. (Qingdao, China). MWCNT with average diameter of 15 nm and length of 2 μm was provided by Jiangsu Hengqiu Carbon Materials Inc. (Suzhou, China).

Prior to the extrusion foaming, MWCNT, mGr and lignin powders were dried in vacuum over night, and the compounding of PS/Lignin and PS/carbonaceous fillers was carried out using a twin-screw extruder (Leistritz ZSE-27, Nuernberg, German; L is for length and D is for diameter, L/D = 40:1; D = 27 mm) with the content of 10 wt% MWCNT, 10 wt% and 50 wt% lignin, respectively. Extrusion foaming of PS and PS composites was carried out in the same twin-screw extruder. The blowing agent CO_2_ was firstly cooled and pressurized in a Telydyne ISCO Model 500D (Lincoln, NE, USA) syringe pump, then pumped into the twin-screw extruder through a gas/liquid injection port located at L/D = 16 from the hopper of the extruder.

### 2.2. Morphology Characterization

The cell morphology of the foams were characterized by using scanning electron microscopy (SEM, type S-4700, JEOL, Tokyo, Japan). The fractured surface of PS foams were obtained by immersing samples in liquid nitrogen and spayed with gold before SEM examination. 

The dispersions of multi-walled carbon tubes, microcrystalline graphite and lignin were observed by transmission electron microscopy (TEM, JEM-100 CX II, 300 kV, JEOL, Tokyo, Japan).

### 2.3. Mechanical Properties

The dynamic thermal mechanical analysis was conducted using dynamic thermal mechanical analyzer (DMA, type Q-800, TA Instruments, New Castle, DE, USA). Samples were obtained by hot pressing. The sample size was cut to small plate with a scale of 30 × 10 × 2 mm^3^. The mode was single cantilever. The temperature range was from 40 to 150 °C. The heating rate and frequency were set to 3 °C/min and 1Hz, respectively.

Samples were cut into 5 × 5 cm^2^ square sample blocks and their thickness were measured after grinding. Three samples were made for each group and compressive strength test was carried out. Using Instron 5966, the compression test was carried out at the compression speed of 1.000 mm/min. The modulus of each sample was calculated at the pressure of 50% compression deformation. Finally, three groups of numerical average values were obtained.

### 2.4. Sound Absorption Property

Absorption was tested using a Bruel and Kjaer, four-microphone small standing wave tube [20] (Type: 4206-T, the length is 50 cm and the diameter is 29 cm). The effective sound wave was measured in the range from 500 to 6000 Hz at 25 °C. The thickness of all samples was 5 mm and the diameter was 29 cm. Three specimens were tested to calculate the average value.

### 2.5. Density Test

The weight (*m*) of samples was measured by the electronic scales (FA1104N, Shanghai, China). The initial water volume (*V*_0_) and the volume (*V*) after the samples needled into the water were measured by the measuring cylinder. The density equals quality divided by the volume that was *V* minus *V*_0_. At least five specimens for each sample were tested and the average value was calculated.
(1)$$\rho =\frac{m}{V-V_{0}}$$

### 2.6. Cell Morphology

After the foaming sample was placed in liquid nitrogen for 2 min, the chips were quickly broken and sprayed. The bubble distribution and the morphology of the brittle fracture surface were observed by scanning electron microscope. The SEM pictures obtained were processed and analyzed by Gatan Digital Micrograph software (Gatan, Pleasanton, CA, USA), and the average bubble size and cell density of the foams were obtained. The cell density (*N*_0_) is the number of cells per cubic centimeter and can be calculated by:
(2)$$N_{0}={\left(\frac{n}{A}\right)}^{\frac{3}{2}}\frac{\rho_{unfoamed}}{\rho_{foam}}$$
where *n* is the number of the cells in a single SEM picture and the *A* is the area of the SEM picture. *ρ_foam_* is the foam density and *ρ_unfoamed_* is the bulk density of PS composites. The average cell diameter (*D*) can be calculated by:
(3)$$D=\frac{\sum n_{i}d_{i}}{\sum n_{i}}$$
where *d_i_* is the diameter of each cell and *n_i_* is the number of cells with the diameter *d_i_* in the SEM pictures.

### 2.7. Thermogravimetry Analysis

Samples less than 10 mg were cut from each sample and heated from 40 to 800 at a heating rate of 20 °C/min in N_2_ atmosphere (TGA Q5000, TA, New Castle, DE, USA).

## 3. Results and Discussion

### 3.1. Cell Morphology

Figure 1 shows TEM images of various PS composites taken by the freezing section method. It was obvious that lignin with low content (10 wt%) was well dispersed in the PS matrix, presenting globular agglomerates at the micron and submicron scales (Figure 1a). Considering the large content of aromatic structure in lignin skeleton, the lignin has excellent compatibility with the PS matrix [21]. With the increase of the lignin content, the size of lignin aggregations was gradually increased, and the composite sample with 50 wt% lignin content showed rich lignin-phase separation (Appendix A). Figure 1b,c depicted that both the MWCNT and the mGr particulates were well dispersed in the PS matrix at low concentration (1 wt%), illustrating the 1D or 2D orientation of the nanoparticles. The dispersion and the orientation of fillers strongly influenced the hetero-nucleation efficiency during the extrusion foaming process.

According to classical nucleation theory [22,23,24], the heterogeneous nucleation rate (*N*_het_) can be expressed as:
(4)$$N_{\mathrm{het}}=f_{\mathrm{het}}C_{\mathrm{het}}\exp\left(-\Delta G_{\mathrm{het}}^{*}/kT\right)$$
where *f*_het_ is the frequency factor of gas molecules joining the nucleus, *C*_het_ is the concentration of heterogeneous nucleation sites, k is the Boltzmann’s constant, *T* is the temperature, and $\Delta G_{\mathrm{het}}^{*}$ is the Gibbs free energy associated with the formation of a nucleus. $\Delta G_{\mathrm{het}}^{*}$ is related to the interfacial tension (*γ*) and the difference (Δ*P*) between the pressure inside the critical nuclei and around the surrounding liquid as:
(5)$$\Delta G_{\mathrm{het}}^{*}=\frac{16\pi \gamma }{3(\Delta P^{2})}f(\theta ,\omega )$$
where *f*(*θ*,*ω*) represents the corrected factor of heterogeneous nucleation, which is a function of the polymer-gas-particle contact angle (*θ*) and the relative curvature *ω* of the nucleate surface to the critical radius of the nucleated phase. Qualitatively, a small contact angle and a large surface curvature offer a higher reduction of critical energy, and consequently an increase in the nucleation rate [25].

We can obtain foamed boards (12 cm width × 1 cm thickness) by using the extrusion foaming with a slit die, which is expected to be easy for industrial and mass-scale production. The upper images in Figure 2 and Figure 3 illustrated that the cross section of various foams were relatively uniform and large foamed boards can be easily produced at the industrial scale. The cell morphology of the PS/lignin and PS/carbonaceous filler composite foams prepared by supercritical CO_2_ foaming is shown in Figure 2 and Figure 3, respectively. It is indicated that the these are closed porosity foams. For both lignin and carbonaceous nanoparticles (MWCNT and mGr), the cell size became smaller and the cell density became higher with increasing the content of fillers, which contributed to the hetero-nucleation effect. Figure 2 displays that the average cell diameter increased from 175 μm to 413 μm after adding 10% lignin. With the increase of lignin content, the average cell size decreased gradually from 414 to 141 μm, and the cell density increased from 1.14 × 10^5^ to 1.34 × 10^6^ cells/cm^3^. The cell size of the foam sample with 40 wt% lignin reduced greatly, due to the fact that the lignin domain is spherical so that the nucleation efficiency is not significant until the content and phase size of lignin are sufficient for hetero-nucleation effect. It is worth mentioning that the extrusion foaming process was still successful when lignin was added to the 50 wt% (Figure 2d), which has not been reported before. The advantage of using lignin as a hetero-nucleation agent is that it is a natural product and abundant resource obtainable at very low cost.

Meanwhile, the composite foams with MWCNT and mGr nanoparticles exhibited similar cell size (100~350 μm) and cell density (1.38 × 10^5^~2.63 × 10^6^ cells/cm^3^) as shown in Figure 3. Both MWCNT and mGr displayed excellent hetero-nucleation efficiency at low contents (0.5 wt%) due to their one-dimensional tubular and two-dimensional layered structure [26,27]. The carbonaceous fillers are CO_2_-philic and the gas bubble can be easily nucleated at the edge of nanoparticles, so the increase of filler content is attributed to higher nucleation and cell growth rate. The density of different PS composite foams is summarized in Appendix A.

### 3.2. Sound Absorption Property

Generally, a sound wave is reflected at the solid-gas interface and dissipated in the cellular structure on the propagation route. The sound absorption efficiency of PS composite foams can also be influenced by the modulus of the polymer, the cell size and cell density. Figure 4 compared the sound absorption performance of different PS composite foams. All the samples showed a tremendous sound absorption coefficient at normal incidence in a range of certain frequency. The pure PS foam showed an obvious peak of sound absorption coefficient at normal incidence around 5100 Hz, whose value could reach as high as 0.97. With the addition of filler, the peak of sound absorption coefficient at normal incidence moved to low frequency zone firstly, and then shifted to high frequency zone gradually. For example, the peak position of sound absorption coefficient at normal incidence of PS/lignin composite foams was shifted from 4216 Hz (10 wt% lignin, 368 μm cell size) to 5552 Hz (50 wt% lignin, 140.6 μm cell size), while all the maximum values were higher than 0.90. This can be explained by the different cell size and cell density that mainly dominated the dissipation of reflect sound wave. Despite the modulus of the polymer being related to the surface reflection, the sound absorption coefficient at normal incidence has little effect on the modulus because of the very rigid state of PS and its composites at room temperature. With the increase of the filler content, the filler as nucleation agent promoted the formation of the cell, the cell size decreased and the cell density increased. Meanwhile, the absorption range of PS composite foams became broader than pure PS foam. When the sound wave was incident into the PS composites, more sound energy can be dissipated in the polymer. Moreover, the propagation distance of the sound wave was increased, which further enhanced the internal dissipation of sound energy. This corroborated with the previous SEM results. The sound absorption coefficient at normal incidence was composed of sharp, narrow peaks, meaning the foams absorbed sound selectively in a relatively narrow band, suggesting the use of this material for applications requiring a narrow acoustic absorption bands.

The acoustic energy dissipation of materials was related to the loss modulus and tan *δ* of the materials [13]. In this work, the loss modulus measured by DMA is used to characterize the sound absorption and noise reduction performance of PS composite foams, and the energy dissipation performance of the PS foam material can be represented. The temperature dependence curves of storage modulus, loss modulus and tan *δ* were shown in Figure 5. It can be found that the storage modulus of PS composites has been greatly improved compared with pure PS in the whole temperature range after adding different fillers. This indicated that the stiffness of foamed materials has been greatly improved by adding both lignin and carbonaceous nanoparticles. The loss modulus of PS composite also improved in comparison with that of pure PS, which indicated that PS composite foams can restrain more mechanical vibration and dissipate more acoustic energy in the process of sound propagation. With the increase of lignin content, tan *δ* of PS/lignin composites decreased slightly. This phenomenon may be due to the phase separation in PS/lignin composite, and the heterogeneous domain will restrain the movement of the polymer chain [28,29,30].

### 3.3. Mechanical Property

Figure 6 plots the specific compression strength (divide compressive stress by foam density) of PS composite foams at the 50% strain. As shown in Figure 5a, this indicated that the specific compressive strength gradually increased with the increase of MWCNT or mGr content, and the specific compressive strength have been greatly improved at low filler content (1 wt%). It is because the MWCNT has a super-high aspect ratio, and the mGr also has a larger surface area and aspect ratio, that it provided very low percolation value of the nanoparticles [31,32]. On the other side, the specific compressive strength of PS/Lignin composite foam decreased gradually with the increase of lignin content (Figure 5b). Considering the intrinsic hyper-branch structure and low molecular weight of lignin molecules, the lignin is much brittler than pure PS. Besides the phase separation of PS/lignin composites, the mechanical property PS/lignin composite foams is reasonably weaker than the pure PS foam. This phenomenon has been previously reported in other polymer/lignin composites [33].

### 3.4. Thermal Stability

Table 1 summarizes the thermal decomposition behaviors of different PS composite foams. It can be seen that MWCNT and mGr have little effect on the thermal decomposition temperature of PS composite foams, while PS/lignin composite foams obtained higher degradation temperatures and char residues. The decomposition temperature of 50 wt% weight loss and the char weight increased with the increase of lignin content, which indicated that lignin can achieve flame retardant effects in PS composite foams [34].

## 4. Conclusions

In summary, a series of PS composite foams were prepared by continuous extrusion foaming assisted by supercritical CO_2_. Lignin was successfully compounded in a PS matrix with high content and foamed with various cell structures on the basis of the hetero-nucleation effect. Compared with PS/MWCNT or PS/mGr composite foams, despite the hetero-nucleation effect of lignin being weaker than that of carbonaceous nanoparticles the PS/lignin composite foam exhibited variable sound absorption coefficients at normal incidence in ranges of much broader sound wave frequency. This is ascribed to the cell size and cell density being essential for reflection and dissipation of sound wave propagation. Moreover, lignin with high content tented to agglomerate and separate from the PS matrix, which could increase the energy loss of the sound wave. The compressive strength of PS composite foams was tested to confirm the dispersion of lignin and carbonaceous particles in the PS matrix. The lignin also showed a fire-retardant effect in the PS composite foam. In view of the need for industrial polymer foams for sound absorption applications, lignin is an alternative option for producing low-cost and environmentally benign polymer foams.

## Figures and Tables

**Figure 1 polymers-11-00106-f001:**
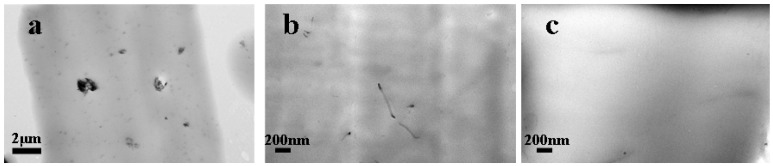
The transmission electron microscope (TEM) images of particles dispersed in polystyrene (PS). (**a**) 10% Lignin; (**b**) 0.2% multiwall carbon nanotube (MWCNT); (**c**) 1% mGr.

**Figure 2 polymers-11-00106-f002:**
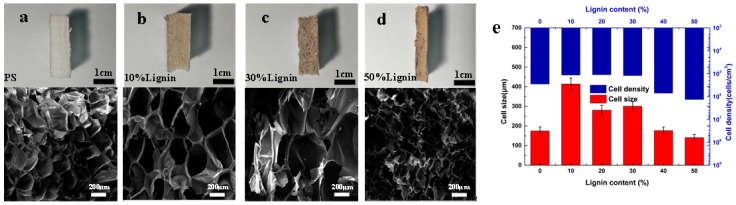
The cross-section view of extruded PS/lignin composite foams and representative cell morphology observed by scanning electron microscope (SEM) (the magnification ratio was 100 times). (**a**) pure PS; (**b**) 10 wt% lignin; (**c**) 30 wt% lignin; (**d**) 50 wt% lignin; (**e**) the cell size and cell density of PS/lignin composite foams.

**Figure 3 polymers-11-00106-f003:**
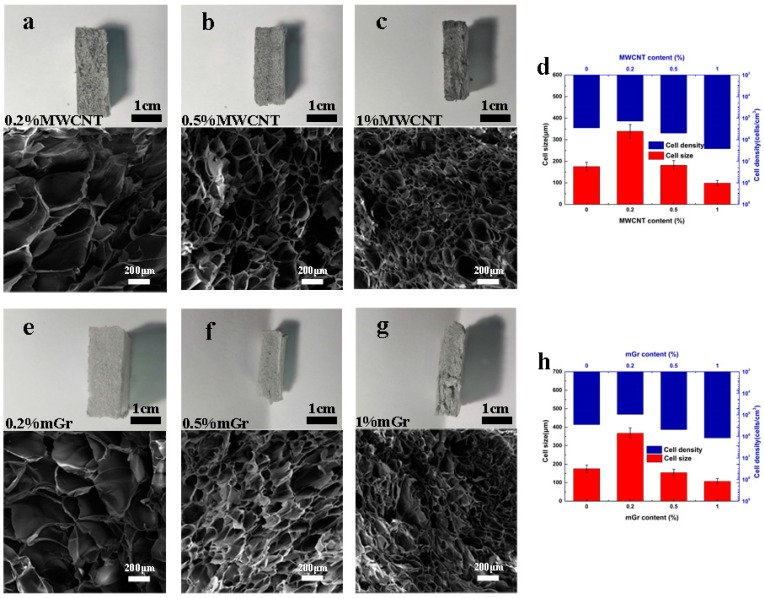
The cross-section view of extruded PS/carbonaceous fillers composite foams and representative cell morphology observed by SEM (the magnification ratio was 100 times). (**a**) 0.2 wt% MWCNT; (**b**) 0.5 wt% MWCNT; (**c**) 1 wt% MWCNT; (**d**) the cell size and cell density of PS/MWCNT composite foams; (**e**) 0.2 wt% mGr; (**f**) 0.5 wt% mGr; (**g**) 1 wt% mGr; (**h**) the cell size and cell density of PS/mGr composite foams.

**Figure 4 polymers-11-00106-f004:**
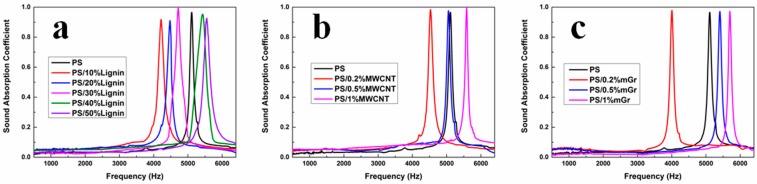
Sound absorption property of PS composite foams: (**a**) PS/lignin; (**b**) PS/MWCNT; (**c**) PS/mGr.

**Figure 5 polymers-11-00106-f005:**
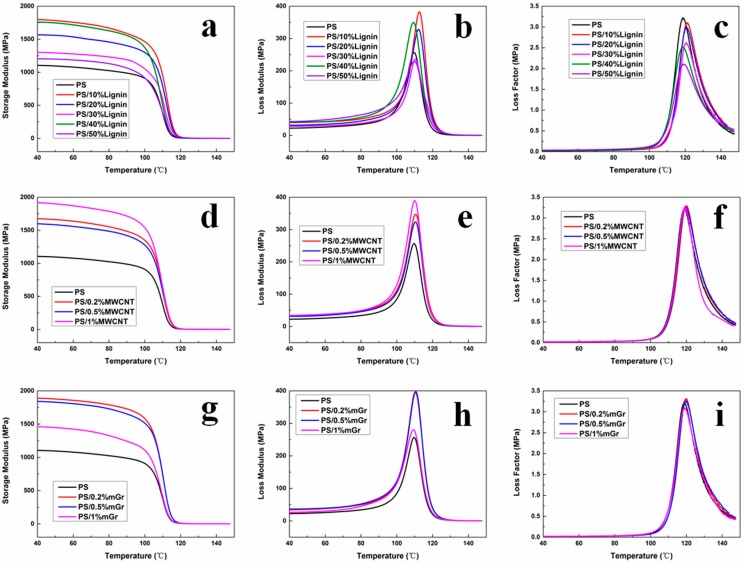
Dynamic thermal mechanical analyzer (DMA) results of different PS composites: (**a**) storage moduli (*E*′) of PS/lignin composites; (**b**) loss moduli (*E*″) of PS/lignin composites; (**c**) tan *δ* of PS/lignin composites; (**d**) storage moduli (*E*′) of PS/MWCNT composites; (**e**) loss moduli (*E*″) of PS/MWCNT composites; (**f**) tan *δ* of PS/MWCNT composites; (**g**) storage moduli (*E*′) of PS/mGr composites; (**h**) loss moduli (*E*″) of PS/mGr composites; (**i**) tan *δ* of PS/mGr composites.

**Figure 6 polymers-11-00106-f006:**
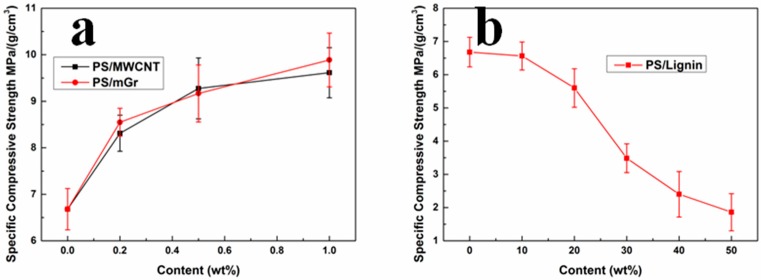
Specific compressive strength of different PS composite foams: (**a**) PS/MWCNT and PS/mGr; (**b**) PS/lignin.

**Table 1 polymers-11-00106-t001:** The decomposition temperature of PS composite foams.

Sample	10 wt% Degradation (°C)	50 wt% Degradation (°C)	Char Weight (%)
Pure PS	397.14	422.72	0.12
PS/0.2% MWCNT	398.10	423.20	0.32
PS/0.5% MWCNT	398.09	423.35	0.53
PS/1.0% MWCNT	397.96	423.87	1.15
PS/0.2% mGr	397.97	422.90	0.33
PS/0.5% mGr	396.92	423.00	0.76
PS/1.0% mGr	392.87	419.45	1.37
PS/10% Lignin	400.92	423.87	1.74
PS/20% Lignin	401.61	425.06	2.72
PS/30% Lignin	401.46	428.21	5.96
PS/40% Lignin	394.22	435.38	11.47
PS/50% Lignin	388.73	435.65	14.49

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
