# Peer review of "Extrusion Foaming of Lightweight Polystyrene Composite Foams with Controllable Cellular Structure for Sound Absorption Application"

_polymers, 2019, doi:10.3390/polym11010106_

Round 1
Reviewer 1 Report
Generally, the manuscript is well written and well structured. However, some points related to sections 2.5 and 3.2. related to “Sound absorption property” can be considered to improve the manuscript and to publish this paper in the journal. The following comments are in general minor changes:
The sound absorption coefficient of an absorbing material depends on the frequency, the incidence angle of the sound and on the way the material is mounted or installed. There are two main types of methods for determining the absorption coefficient of acoustic materials: the reverberation time method for diffuse field; and the impedance tube method for normal incidence.
In the manuscript the authors say “absorption coefficient”, however that has to be corrected and they must specified that is the “sound absorption coefficient at normal incidence”.
In the discussion of the maximum absorption position according to the cell size of the manufactured samples, the authors should also contemplate the discussion considering that the cavities behave like Helmholtz micro-perforated resonators.
Author Response
Dear editors,
Thank you very much for evaluating our manuscript. Based on the opinions of the reviewers, we have revised the manuscript as follows.
Reviewer: 1
Generally, the manuscript is well written and well structured. However, some points related to sections 2.5 and 3.2. related to “Sound absorption property” can be considered to improve the manuscript and to publish this paper in the journal. The following comments are in general minor changes:
The sound absorption coefficient of an absorbing material depends on the frequency, the incidence angle of the sound and on the way the material is mounted or installed. There are two main types of methods for determining the absorption coefficient of acoustic materials: the reverberation time method for diffuse field; and the impedance tube method for normal incidence.
In the manuscript the authors say “absorption coefficient”, however that has to be corrected and they must specified that is the “sound absorption coefficient at normal incidence”.
Response: Thank you very much! We have corrected the expression of absorption coefficient in the article.
In the discussion of the maximum absorption position according to the cell size of the manufactured samples, the authors should also contemplate the discussion considering that the cavities behave like Helmholtz micro-perforated resonators.
Response: Thank you very much! The Helmholtz resonator frequency is very high and it's not in the range of resonance frequency. Sound absorption coefficient should be independent of Helmholtz resonator effect.

Reviewer 2 Report
The article “ Extrusion Foaming of Lightweight Polystyrene Composite Foams with Controllable Cellular Structure for Sound Absorption Application” by Yanpei Fei et al, reports on the synthesis of a series of PS composite foams, prepared by continuous extrusion foaming assisted by supercritical CO2 for sound absorption. The cell size cell density which are key to determining the sound absorption coefficient were modified by the bio-resource lignin. This was the first time lignin was employed to control the cellular structure on basis of hetero-nucleation effect.
Multiwall Carbon Nanotubes (MWCNT- please define this in the text) and micro-graphite particles were also employed as nucleation agents during the foaming process. They were found to be more efficient on the hetero-nucleation effect.
General remarks
The sound absorption properties were measured using an impedance (Kundt) tube using four microphones in the range from 500 to 6000 Hz at 25°C. The absorption coefficient was composed of sharp, narrow peaks, meaning the foams absorbed sound selectively in a relatively narrow band (this should be mentioned in the text). The use of this material for applications requiring a narrow acoustic absorption bands should be mentioned.
Please cite appropriately
Ancuţa-Elena Tiuc and Horaţiu Vermeşan and Timea Gabor and Ovidiu Vasile, Improved Sound Absorption Properties of Polyurethane Foam Mixed with Textile Waste, Energy Procedia, Volume 85, 2016, Pages 559-565, ISSN 1876-6102, https://doi.org/10.1016/j.egypro.2015.12.245.
The viscoacoustics parameters (porosity, tortuosity …) influencing acoustic absorption of the foams were not determined for a more detailed study of the geometric parameters being varied by adding the foaming agents. Please mention that these are closed porosity foams.
For the determination of these parameters using the Kundt tube, See: Mohamed Ben Mansour, Erick Ogam, Ahmed Jelidi, Amel Soukaina Cherif, Sadok Ben Jabrallah, Influence of compaction pressure on the mechanical and acoustic properties of compacted earth blocks: An inverse multi-parameter acoustic problem, Pages 128-135. Please cite appropriately.
The loss modulus and tan (delta) were measured using Dynamic mechanical analysis (DMA – please give the meaning of DMA in the text and provide the references of the name of the equipment employed and place of manufacture)
DMA
Cell Morphology was analyzed using Digital Micrograph software (the microscope brand and place of manufacture, the details of the software should be mentioned).
Detailed remarks
Page 2 Materials and methods, please mention city and country of origin for all the equipment and samples employed.
Page 2 Line 63, what are L and D.
Page 2 Line 77 30×10×2 mm should read 30×10×2 mm^3
Page 2 Line 80 5×5 cm -> 5×5 cm^2. Please correct the sentence “ … their thickness was measured …“ to “… their thickness were measured …”.
Page 2, 2.4. Sound Absorption Property, please give the length, the diameter of the kundt tube.
Page 3 Line 95 Please number all equations
Page 3 Line 102 The equation is tiny, please enlarge.
Page 3 Line 115 Fig. 1 The TEM of particles …, did you mean SEM?
Page 3 Line 121, (Fig. S1). (Fig. S1) does not exist.
Page 3 Fig. 2, Fig 3, please enlarge and improve the quality of the SEM images and please mention what is found on the upper panels clearly (the cross sections are in exploitable).
Page 5 Line 144 there is a problem with this equation.
Page 5 Fig 4. Please enlarge the plots.
Page 6 Fig 5. Please enlarge the plots.
Author Response
Dear editors,
Thank you very much for evaluating our manuscript. Based on the opinions of the reviewers, we have revised the manuscript as follows.
Reviewer: 2
General remarks
The sound absorption properties were measured using an impedance (Kundt) tube using four microphones in the range from 500 to 6000 Hz at 25°C. The absorption coefficient was composed of sharp, narrow peaks, meaning the foams absorbed sound selectively in a relatively narrow band (this should be mentioned in the text). The use of this material for applications requiring a narrow acoustic absorption bands should be mentioned.
Response: Thank you very much!We have mentioned the foams absorbed sound selectively in a relatively narrow band and the applications requiring a narrow acoustic absorption bands in the article.
Please cite appropriately
Ancuţa-Elena Tiuc and Horaţiu Vermeşan and Timea Gabor and Ovidiu Vasile, Improved Sound Absorption Properties of Polyurethane Foam Mixed with Textile Waste, Energy Procedia, Volume 85, 2016, Pages 559-565, ISSN 1876-6102, https://doi.org/10.1016/j.egypro.2015.12.245.
Response: Thank you very much! We quoted this document in the article.
The viscoacoustics parameters (porosity, tortuosity …) influencing acoustic absorption of the foams were not determined for a more detailed study of the geometric parameters being varied by adding the foaming agents. Please mention that these are closed porosity foams.
For the determination of these parameters using the Kundt tube, See: Mohamed Ben Mansour, Erick Ogam, Ahmed Jelidi, Amel Soukaina Cherif, Sadok Ben Jabrallah, Influence of compaction pressure on the mechanical and acoustic properties of compacted earth blocks: An inverse multi-parameter acoustic problem, Pages 128-135. Please cite appropriately.
Response: Thank you very much! We have mentioned that these are closed porosity foams and quoted this document in the article.
The loss modulus and tan (delta) were measured using Dynamic mechanical analysis (DMA – please give the meaning of DMA in the text and provide the references of the name of the equipment employed and place of manufacture)
Response: Thank you very much! We have introduced the DMA in detail in 2.3 mechanical properties.
Cell Morphology was analyzed using Digital Micrograph software (the microscope brand and place of manufacture, the details of the software should be mentioned).
Response: Thank you very much! We have completed the software information.
Detailed remarks
Page 2 Materials and methods, please mention city and country of origin for all the equipment and samples employed.
Page 2 Line 63, what are L and D.
Response: Thank you very much! We have added the information of all the equipment, samples employed and the meaning of L and D.
Page 2 Line 77 30×10×2 mm should read 30×10×2 mm^3
Page 2 Line 80 5×5 cm -> 5×5 cm^2. Please correct the sentence “ … their thickness was measured …“ to “… their thickness were measured …”.
Response: Thank you very much! We have corrected the wrong expression.
Page 2, 2.4. Sound Absorption Property, please give the length, the diameter of the kundt tube.
Response: Thank you very much! We have added the length and the diameter of the kundt tube.
Page 3 Line 95 Please number all equations.
Response: Thank you very much! All equations have been numbered.
Page 3 Line 102 The equation is tiny, please enlarge.
Response: Thank you very much! We have enlarged the equation.
Page 3 Line 115 Fig. 1 The TEM of particles …, did you mean SEM?
Response: Thank you very much! Fig.1 is the TEM photographs.
Page 3 Line 121, (Fig. S1). (Fig. S1) does not exist.
Response: Thank you very much! Fig. S1 was supplied in supplementary information.
Page 3 Fig. 2, Fig 3, please enlarge and improve the quality of the SEM images and please mention what is found on the upper panels clearly (the cross sections are in exploitable).
Response: Thank you very much! We have enlarged and improved the quality of the SEM images and mentioned the upper panels in the article.
Page 5 Line 144 there is a problem with this equation.
Response: Thank you very much! We have corrected the equation.
Page 5 Fig 4. Please enlarge the plots.
Page 6 Fig 5. Please enlarge the plots.
Response: Thank you very much! We have enlarged the plots.

Round 2
Reviewer 2 Report
Line 63 German should read Germany. Please give the cities/towns where the equipment was manufactured. For example Leistritz ZSE-27, Nuremberg Germany. Nuremberg Nuremberg NurembergNuremberg